# Interplay between Protein Kinase C Epsilon and Reactive Oxygen Species during Myogenic Differentiation

**DOI:** 10.3390/cells12131792

**Published:** 2023-07-05

**Authors:** Giulia Pozzi, Valentina Presta, Elena Masselli, Giancarlo Condello, Samuele Cortellazzi, Maria Luisa Arcari, Cristina Micheloni, Marco Vitale, Giuliana Gobbi, Prisco Mirandola, Cecilia Carubbi

**Affiliations:** 1Department of Medicine and Surgery (DiMeC), University of Parma, Via Gramsci, 14, 43126 Parma, Italy; giulia.pozzi@unipr.it (G.P.); valentina.presta@unipr.it (V.P.); elena.masselli@unipr.it (E.M.); giancarlo.condello@unipr.it (G.C.); marialuisa.arcari@unipr.it (M.L.A.); cristina.micheloni@unipr.it (C.M.); marco.vitale@unipr.it (M.V.); cecilia.carubbi@unipr.it (C.C.); 2Italian Foundation for Research in Balneotherapy (FoRST), 00198 Rome, Italy

**Keywords:** PKC epsilon, SOD2, antioxidant, reactive oxygen species, skeletal muscle, myogenesis

## Abstract

Reactive oxygen species (ROS) are currently recognized as a key driver of several physiological processes. Increasing evidence indicates that ROS levels can affect myogenic differentiation, but the molecular mechanisms still need to be elucidated. Protein kinase C (PKC) epsilon (PKCe) promotes muscle stem cell differentiation and regeneration of skeletal muscle after injury. PKCs play a tissue-specific role in redox biology, with specific isoforms being both a target of ROS and an up-stream regulator of ROS production. Therefore, we hypothesized that PKCe represents a molecular link between redox homeostasis and myogenic differentiation. We used an in vitro model of a mouse myoblast cell line (C2C12) to study the PKC–redox axis. We demonstrated that the transition from a myoblast to myotube is typified by increased PKCe protein content and decreased ROS. Intriguingly, the expression of the antioxidant enzyme superoxide dismutase 2 (SOD2) is significantly higher in the late phases of myogenic differentiation, mimicking PKCe protein content. Furthermore, we demonstrated that PKCe inhibition increases ROS and reduces SOD2 protein content while SOD2 silencing did not affect PKCe protein content, suggesting that the kinase could be an up-stream regulator of SOD2. To support this hypothesis, we found that in C2C12 cells, PKCe interacts with Nrf2, whose activation induces SOD2 transcription. Overall, our results indicate that PKCe is capable of activating the antioxidant signaling preventing ROS accumulation in a myotube, eventually promoting myogenic differentiation.

## 1. Introduction

Generation of reactive oxygen species (ROS) is a ubiquitous phenomenon in eukaryotic cell life that leads to the production of signaling molecules, which regulate a plethora of physiological processes. However, a univocal description of ROS signaling functions is arduous given that their biological effects may radically change according to their intracellular type, level, and subcellular localization that in turn are tissue- and cell-specific.

In skeletal muscle, ROS production occurs both at rest and during contractile activity, promoting complex and divergent effects, ranging from positive to detrimental [1]. 

Skeletal muscle fibers are polynuclear cells that, during embryonic development, as well as during muscle regeneration, are formed in a complex process involving muscle satellite cell (mSC) activation, fusion of proliferating myoblast cells into myotubes, and further maturation of myotubes to muscle fibers [2]. Myogenesis is accompanied by an intensive metabolic remodeling that leads to changes in ROS synthesis. Increasing evidence indicates that ROS can affect myogenic differentiation, albeit with somewhat contradictory data. Indeed, most authors reported that ROS can cause inhibition of myogenic differentiation [3,4,5,6,7,8] by (i) NF-kB activation, which in turn causes a reduction in MyoD levels [9]; (ii) increased cyclin D1 transcription and cell proliferation [10]; and (iii) enhanced expression of YY1, a transcriptional repressor of myogenic genes [11]. Moreover, it has also been demonstrated that higher amounts of ROS might target the mitochondria and mitochondrial DNA, inducing the block of myogenesis [12,13]. However, pro-oxidative signaling is crucial for mSC activation at the injury site of a regenerating skeletal muscle [8,14,15,16,17] and it has been demonstrated that the abrogation of physiological ROS signaling hampers differentiation of mSC [18]. Furthermore, it has been reported that the transition from myoblasts to myotubes is accompanied by a reduction in intracellular ROS [19,20,21,22].

Muscle differentiation is also accompanied by changes in the activity of the intracellular antioxidant system enzymes, which obviously affects the overall intracellular ROS [8]. Specifically, the reduction in antioxidant systems seems to be related to inhibition of muscle regeneration and of mSC recruitment [6,23,24,25]. The major class of antioxidant enzymes, which protects against primary ROS, O_2_^−^, is superoxide dismutase (SOD). SOD1 and SOD3 require Cu–Zn as a cofactor and are located in the cytosol and in the extracellular space, respectively. SOD1 appears to be involved in age-related muscle atrophy and contractile force. Indeed, elevated levels of oxidative stress, a loss of muscle contraction, and accelerated muscle atrophy have been reported in SOD1^−/−^ mice [26,27,28]. However, these effects are independent from the muscle fibers *per se*, as demonstrated by muscle-specific SOD1 deficiency, but are rather primarily related to impaired redox signaling in peripheral nerves and neuromuscular junctions in SOD1^−/−^ mice [29,30].

SOD2 (or MnSOD) has a mitochondrial localization and requires Manganese (Mn) as a cofactor. Mn influx and the dynamic expression of Mn-transporting proteins are crucial factors to sustain skeletal muscle cell differentiation [31]. Indeed, a concomitant increase in Mn influx and SOD2 levels in murine primary myoblasts during myogenic differentiation has been reported. Additionally, murine models of SOD2 deficiency displayed mitochondrial dysfunction [32], a ROS increase [33], and a trend of altered expression of myogenesis transcriptional regulators (*Mef2c*, *Myog*, and *Myod*), indicating that a loss of SOD2 may impact on muscle myogenesis [34].

Among the molecular pathways involved in myogenesis, we previously demonstrated that protein kinase C epsilon (PKCe) is finely tuned during mSC differentiation, promoting myogenic differentiation and regeneration of skeletal muscle after injury [35,36]. PKCe is a serine/threonine kinase of the protein kinase C family, which encompasses several protein isoforms (PKCs). PKCs are involved in a variety of pathways that regulate cell growth, differentiation, apoptosis, and tumorigenicity [37,38,39,40,41,42,43,44,45,46,47,48,49]. 

Recent data describe a tissue-specific role of PKCs in redox biology [50,51]. Most of the literature supports a direct activation of different PKC isoforms by ROS [52,53,54]. For instance, in mouse embryonic fibroblasts, oxidative stress triggers translocation, and subsequent activation, of PKC alpha, beta, delta, and epsilon isoforms [54,55]; in arterial smooth muscle cells, an increase in mitochondrial ROS leads to the activation of total PKC and PKCe; similarly, the external addition of H_2_O_2_ is able to increase PKCe activity in pulmonary arteries [54]. On the other hand, PKC-induced signaling is capable of affecting ROS. Indeed, the well-documented PKCe cardioprotective function is mediated by a direct interaction between the kinase and mitochondrion, which in turn affects ROS release [54]. Additionally, PKCs are able to phosphorylate and activate NOX2, therefore inducing ROS in phagocytes [56,57], aortic endothelial cells, platelets [58], and renal mesangial cells [59]. Moreover, in a cancer model, we demonstrated that PKCe inhibition increases ROS in acute myeloid leukemia cells, phenocopying SOD2 inhibition [51]. Furthermore, SOD2 appears to be down-stream of PKCe signaling in non-muscle cells: (i) in a lung adenocarcinoma cell line, TPA (12-O-tetradecanoylphorbol-13-acetate) up-regulated PKCe- and PKCalpha-dependent transcriptional pathways to induce SOD2 expression [60] and (ii) in primary human hippocampal neuron cultures, the loss of PKCe reduced SOD2, resulting in oxidative stress, and these effects were reverted by the PKCe activator bryostatin in a murine model of Alzheimer’s disease [61].

Based on the observations that ROS and PKCe mutually affect their expression and function and that both are involved in myogenic differentiation, in this study, we sought to decipher the redox–PKCe crosstalk in myogenic differentiation.

## 2. Materials and Methods

### 2.1. Cell Culture

C2C12 are precursor cells in adult mouse skeletal muscle tissue that regenerate muscle tissue after trauma, with an excellent ability for growth and differentiation, and are therefore a well-established cell model for in vitro skeletal muscle differentiation. To maintain myoblast proliferation, we cultured a mouse myoblast C2C12 cell line (purchased from the American Type Culture Collection, ATCC, Rockville, MD, USA) in a growth medium (GM). GM was a Dulbecco’s modified Eagle’s medium (DMEM, Corning, New York, NY, USA) supplemented with heat-inactivated 10% fetal bovine serum, FBS (GibcoTM, Thermo Fisher, Waltham, MA, USA), 2 mM of glutamine (Biowest, Nuaillé, France), and 1% antibiotics (penicillin/streptomycin, Biowest, Nuaillé, France). To induce myogenic differentiation in cell cultures reaching 80% of confluence, we replaced GM with a differentiation medium (DM, DMEM supplemented with 2% horse serum, Sigma Aldrich, St. Louis, MO, USA). Cells were maintained in a humidified 5% CO_2_ atmosphere at 37 °C. At each experimental time point, cells were detached using a scraper in ice with cold PBS (phosphate-buffered saline) and collected for a further analysis.

### 2.2. Gene Silencing

#### 2.2.1. Short Hairpin RNA (shRNA) Cell Infection

In specific experiments, knockdown of PKCe was obtained with shRNA gene silencing, as we previously performed [35]. Briefly, we used a pLKO.1 lentiviral vector encoding shRNA against mouse PKCe (shPKCe) (ThermoScientific, Waltham, MA, USA). As a control (shCTRL), we used the MISSION pLKO.1-puro Non-Target shRNA Control Plasmid, containing a shRNA insert unable to target any known genes from any species (Sigma-Aldrich, St. Louis, MO, USA). We produced the shRNA-expressing viruses in 293TL cells, according to standard protocols. The mouse proliferating C2C12 cell line was infected with shPKCe or shCTRL and the infected cells were selected up to 72 h as puromycin-resistant cells by adding puromycin (2 μg/mL) to the differentiation medium.

#### 2.2.2. Small Interfering RNA (siRNA) Cell Transfection

In specific experiments, PKCe and SOD2 silencing was obtained with transfection of 100 nM of specific siRNAs or control siRNA (siCTRL) (Silencer^®^ Select siRNA, Ambion, SOD2 siRNA, Cat. No.: 4390771, Assay ID: s74130; PKCe siRNA, Cat. No.: 4390771, Assay ID: s71699 and s71700; siCTRL Silencer^®^ Select Negative Control, Cat. No.: 4390843). siRNA transfection was performed with a Lipofectamine RNAiMAX reagent (Invitrogen, Waltham, MA, USA) according to the manufacturing protocol. Briefly, lipofectamine was diluted 1:16.6 in Opti-MEM and siRNA was diluted 1:50 in Opti-MEM, then diluted lipofectamine was added to diluted siRNA, gently mixed, and incubated for 5 min at room temperature. The siRNA–lipofectamine mix was added to the differentiation medium (DM) in a ratio of 1:10. Seeded cells, at 60–80% of confluence, were washed with PBS and the transfection mix was added to the cells. Protein and mRNA were then analyzed after 72 h of transfection.

### 2.3. Measurement of ROS

Intracellular ROS were detected with flow cytometry (FCM) and an immunofluorescence (IF) analysis. 

#### 2.3.1. ROS Detection with Flow Cytometry

For the FCM analysis, a CellROX™ Green FCM Assay Kit (Thermo Fisher Scientific, Waltham, MA, USA, Cat. No.: C10492) was used. The cell permeable CellROX^®^ reagent is non-fluorescent but exhibits a strong fluorogenic signal upon oxidation, providing a reliable measure of ROS in live cells. In addition, the common inducer of ROS production, tert-butyl hydroperoxide (TBHP, 200 µM) and the antioxidant N-acetylcysteine (NAC, 0.5 mM), provided by the kit, were used as positive and negative controls, respectively. The test was performed according to the manufacturing protocols and fluorescence was immediately measured with a flow cytometer. Fluorescence emission was collected with a 525/40 BP filter with a CytoFLEX Flow Cytometer (Beckman Coulter, Brea, CA, USA). The FCM data analysis was performed using Kaluza Analysis Software (Beckman Coulter, Brea, CA, USA) and the mean fluorescence intensity was measured for semiquantitation of ROS levels.

#### 2.3.2. ROS Detection with Immunofluorescence

For the immunofluorescence (IF) analysis, a CellROX™ Green Reagent (Thermo Fisher Scientific, Waltham, MA, USA Cat. No.: C10492) was used. The CellROX™ Green Reagent is a novel fluorogenic probe for measuring oxidative stress in live cells. The cell-permeant dye is weakly fluorescent in a reduced state and exhibits bright green fluorescence upon oxidation by ROS and subsequent binding to DNA, with absorption/emission maxima of ~485/520 nm. Briefly, cells were grown in 48-well dishes on a coverslipand, at each specific time point, were incubated with the CellROX^®^ Reagent (final concentration of 5 μM) for 30 min at 37 °C; after incubation, the medium was removed and the cells were washed three times with 1X PBS, fixed with 4% paraformaldehyde for 15 min, permeabilized with 0.5% Triton^®^ X-100 for 10 min, and then stained with a specific antibody anti-Myosin Heavy Chain (see Immunofluorescence paragraph). Samples were observed with a Nikon Eclipse 80i light microscope, and images were acquired at 40× magnification with a Nikon Digital camera DS-U1 and NIS-Element software. The gain and exposure were set in proliferating myoblasts (day 0) and used for each acquisition. The mean cell fluorescence intensity was quantified using ImageJ software and normalized as compared to the mean fluorescence of the background. Moreover, for a more accurate analysis, the corrected total cell fluorescence (CTCF) was also calculated using the following formula: CTCF = Integrated Density—(Area of selected cell X Mean fluorescence of background).

### 2.4. RNA Extraction and Quantitative RT-PCR

For the gene expression analysis, mRNA was extracted with an RNeasy Mini Kit (Qiagen, Frederick, MD, USA), according to the manufacturer’s instructions, and samples with a 260/280 absorbance ratio between 1.9 and 2.0 were stored at −80 °C and further used for reverse transcription. In total, 1 μg of total mRNA was reverse transcribed with a High-capacity RNA-to-cDNA kit (Thermo Fisher Scientific, Waltham, MA, USA) in a mix volume of 20 μL and the cDNA obtained was diluted 1:5 for further real-time quantitative PCR (qPCR). qPCR was performed using a Power-up SYBR Green Master Mix (Thermo Fisher Scientific, Waltham, MA, USA). The cDNAs were amplified with specific primers (Table 1). Mouse beta-glucoronidase (*Gusb*), known to be a good internal control to study mRNA expression in muscular-derived cell lines [35], was used to normalize all results. The primers were selected using an NCBI/primer-blast program (http://www.ncbi.nlm.nih.gov/tools/primer-blast/ (accessed on 1 February 2019)) and were synthesized with an Applied Bio-system (Thermo Fisher Scientific, Waltham, MA, USA). cDNA was amplified in a final mix volume of 10 μL, with 40 run cycles of amplification using a StepOne Real-Time PCR System (Applied Biosystems, Waltham, MA, USA). For the amplification protocol, we used the “default PCR thermal cycling conditions for primers with Tm ≥ 60 °C” reported in the Power-up SYBR Green Master Mix manufacturer protocol. The cDNA from each sample was run in triplicate for each gene of interest, normalized for *Gusb* values, and the fold increase was calculated with the ΔΔCt method (2−(ΔΔCt)).

### 2.5. Analysis of Protein Content with Western Blot

PKC epsilon, SOD2, Nrf-2, and Myogenin protein content was tested with a Western blot. In brief, C2C12 was lysed with a RIPA buffer and total protein was quantified with a BCA protein assay kit (Thermo Fisher Scientific, Waltham, MA, USA), according to the instruction protocol. Forty micrograms of proteins were separated on 10% SDS-polyacrylamide gel electrophoresis, transferred to a nitrocellulose membrane, and incubated with a specific primary antibody diluted in the blotting solution as recommended by the manufacturer protocol. We used anti-PKCe (Merck Millipore, Darmstadt, Germany, Cat. No.: 06-991), anti-SOD2 (Enzo Life Sciences Inc., Farmingdale, NY, USA, Cat. No.: ADI-SOD-111F), anti-Myogenin (Santa Cruz, Dallas, TE, USA, Cat. No.: sc-12732), anti-Nrf2 (clone: D179C, Cell Signaling, Danvers, MA, USA, Cat. No.: 12721), and anti-HSP70 (Sigma-Aldrich, St. Louis, MO, USA, Cat. No.: H5147) as a loading control. The nitrocellulose membranes were then washed and incubated with 1:5000 peroxidase-conjugated anti-rabbit (Thermo Fisher Scientific, Waltham, MA, USA) or 1:2000 peroxidase-conjugated anti-mouse IgG (Sigma-Aldrich, St. Louis, MO, USA). Proteins were resolved with an ECL SuperSignal West Pico Chemiluminescent Substrate Detection System (Thermo Fisher Scientific, Waltham, MA, USA) and densitometric analyses were performed using ImageJ software (by NIH). 

### 2.6. Immunofluorescence 

C2C12 differentiation and PKCe, phospho-PKCe, and SOD2 content was also investigated using the immunofluorescence analysis, as we previously reported [35]. Briefly, cells were grown in 48-well dishes on microscope slide cover glasses. At the indicated time points, cells were washed in PBS and fixed with 4% paraformaldehyde in PBS for 15 min at room temperature and stored in 1X PBS at 4 °C. Samples were permeabilized 3 times with 1% BSA and 0.2% TritonX-100 in 1X PBS for 5 min at room temperature. Then, cells were incubated in a 10% goat serum in PBS for 1 h at room temperature to saturate non-specific binding sites. Samples were incubated for 1.5 h with the primary antibody diluted 1:200 in the 1% goat serum in PBS. PKCe, phospho-PKCe, SOD2, and myosin were detected using an anti-PKCe rabbit antibody (Novus Biologicals, Littleton, CO, USA, Cat. No.: NBP1-30126), anti-PKCe (phospho S729) antibody (Abcam, Cambridge, UK, Cat. No.: ab88241), anti-SOD2 antibody (Enzo Life Sciences Inc., Farmingdale, NY, USA, Cat. No.: ADI-SOD-111F), and anti-Myosin Heavy Chain (MHC) mouse monoclonal antibody (clone MF-20, R&D Systems, Minneapolis, MN, USA, Cat. No.: MAB4470), respectively. Cells were washed in 1×PBS and then incubated with the secondary antibody (Goat anti-Rabbit IgG (H+L) Highly Cross-Adsorbed Secondary Antibody, Alexa Fluor™ 488, Invitrogen, Waltham, MA, USA, Cat. No.: A-11034; Goat anti-Mouse IgG (H+L) Highly Cross-Adsorbed Secondary Antibody, Alexa Fluor™ 546, Invitrogen, Waltham, MA, USA, Cat. No.: A-11030) 1:1000 for 1 h at room temperature. Nuclei were counter-stained using ProLong™ Diamond Antifade Mountant with DAPI (Life Technologies, Waltham, MA, USA, Cat. No.: P36962); fluorescence was observed with a NikonEclipse80i (Nikon, Tokyo, Japan) fluorescent microscope. Images were acquired with a Nikon Camera DS-JMC and analyzed with Nis element F2.30 (Nikon, Japan). Myogenic differentiation was analyzed with the fusion index (number of nuclei in the myotubes/total number of nuclei). The fusion index analysis is reported as a percentage (0% = no detectable fusion event among MF-20 positive cells).

### 2.7. Immunohistochemistry

Formalin-fixed, paraffin-embedded samples of tibialis muscle from wild-type and PKCe-knockout mice (https://www.cancertools.org/experimental-models/151662 (accessed on 1 February 2019)) were kindly provided by Prof. Peter Parker (Francis Crick Institute, London, UK). Sections of 4 µm from paraffin blocks, after horse serum blocking, were incubated with the rabbit anti-SOD2 primary antibody (Enzo Life Sciences Inc., Farmingdale, NY, USA, Cat. No.: ADI-SOD-111F). Immunostaining was carried out with a Vectastain Elite^®^ ABC Universal kit (Vector Laboratories, Newark, CA, USA) and sections were finally counterstained with hematoxylin. Negative controls were treated in parallel without the primary antibody. At least 3 labelled sections from each specimen were observed with a Nikon Eclipse 80i light microscope with 10×, 20×, and 40× magnification and photomicrographs were taken using a connected Nikon Digital camera DS-U1 and NIS-Element software (Nikon).

### 2.8. Immunoprecipitation

A total of 5 × 10^6^ C2C12 cells were lysed in the RIPA buffer, pre-cleared, and incubated with the anti-PKCe antibody (Abcam, Cambridge, UK, 5 μg/mL) or anti-Nrf2 antibody (Cell Signaling, Danvers, MA, USA, 5 μg/mL) overnight at 4 °C. Rabbit anti-human IgG (Sigma Aldrich, St. Louis, MO, USA, 1 μg/mL) was used as a negative control. Samples were incubated with Protein A/G PLUS-Agarose beads (Santa Cruz, Dallas, TE, USA) for 1 h at 4 °C and resolved with SDS-PAGE.

### 2.9. Statistical Analysis

Data are presented as the mean ± SD of at least 3 independent experiments. Statistical analyses were performed using the one-way ANOVA and Tukey test, when applicable. Prism5 software (GraphPad Software Inc., San Diego, CA, USA) was used for all the analyses. A statistically significant difference was reported when *p*-value ≤ 0.05.

## 3. Results

### 3.1. ROS and SOD2 Protein Content Are Modulated during Myogenic Differentiation

C2C12 cells have been used as a well-established model of myogenesis. To assess cell myogenic differentiation in our culture system, we evaluated morphological and molecular features of C2C12 cells treated with DM up to 7 days (Figure 1). As expected, the mRNA expression of the early myogenic differentiation marker *Myf5* increased in the first days of differentiation and then progressively decreased up to 7 days; the mRNA expression of the transcription factor of the intermediate phase of skeletal muscle differentiation, *Myogenin*, peaked at day 2–6; the mRNA expression of the late myogenic differentiation gene *Myf6* increased in the later days of in vitro differentiation (Figure 1A). Myogenic differentiation was also confirmed with the morphological analysis, as C2C12 progressively elongated and fused forming myotubes (Figure 1B,C)), with a consequent significant increase in the fusion index (Figure 1D).

In line with our previous findings [35], the protein content of both total and phosphorylated (i.e., active form) PKCe increased progressively during C2C12 differentiation (Figure 2A–E). 

To investigate a possible redox–PKCe interplay in myogenic differentiation, we tested ROS during myogenesis. After 24 h of DM culturing (day 1), IF showed an increase in ROS, and the subsequent conversion of myoblasts to myotubes was followed by a steady decline in intracellular ROS (Figure 3A–C). Consistently, the FCM analysis showed that total intracellular ROS rapidly increased after 24 h of DM culturing and then significantly decreased after 4 days of differentiation (Figure 3D). Treatment of C2C12 cells with a common inducer of ROS production, TBHP, was capable of inducing an increase in ROS primarily in C2C12 myoblasts (Figure 3E,F; *** *p* < 0.001 + TBHP day 0 vs. + TBHP day 4) while the effects were less evident in differentiated cells. In line with these findings, pre-treatment with the antioxidant NAC produced less effects in myoblasts as compared to myotubes (Figure 3E,F). These results suggest that differentiated myotubes are more resistant to oxidative stress than undifferentiated myoblasts and we asked whether this phenotype could be mediated by increased levels of antioxidant enzymes, which, of note, in skeletal muscle cells, are involved in myogenic differentiation [8].

Specifically, we focused on SOD2, the first main component of the antioxidant system, since previous findings pointed out a role of this enzyme in myogenic differentiation [31,32,33,34], and a functional correlation between SOD2 and PKCe has been proved in non-skeletal muscle models [60,61]. We found that SOD2 protein content increased during C2C12 differentiation (Figure 3G,H) and progressively accumulates in the cytoplasm of myotubes (Figure 3I). These results are consistent with mitochondrial biogenesis and a SOD2 increase reported during primary myoblast differentiation [31]. Moreover, these data suggest that the increase in the antioxidant enzyme, in the late phase of in vitro differentiation, contributes to maintain low ROS in myotubes. 

### 3.2. PKCe Down-Regulation Affects SOD2 Protein Content via Nrf2

As the kinetics of SOD2 expression parallels that of PKCe during C2C12 differentiation, we evaluated whether PKCe could play a role as an up-stream regulator of SOD2 expression. First, we investigated SOD2 protein content in tibialis muscle specimens from the mouse model of PKCe knockout (the PKCe^−/−^ mouse) and compared it with wild-type controls. Despite the heterogeneous content of SOD2 among the fibers of a wild-type mouse, the immunohistochemical analysis revealed a significantly reduced content of the enzyme in the PKCe^−/−^ mouse (Figure 4A). 

This prompted us to investigate whether PKCe could affect SOD2 expression with gene silencing experiments using both shRNA and siRNA technologies in the C2C12 cellular model. We demonstrated that both infection of C2C12 with specific PKCe shRNA and transfection of C2C12 with PKCe siRNA were capable of significantly reducing SOD2 protein content (Figure 4B–F), coupled by a significant decrease in levels of the intermediate marker of myogenic differentiation, Myogenin, as compared to control treatments (Figure 4B–F). This last observation is in line with our previous findings that the PKCe inhibitor peptide reduced C2C12 differentiation [35]. 

Given the effects of PKCe down-regulation on SOD2, we tested ROS after C2C12 transfection with FCM and IF (Figure 5A–C). In fact, we recently demonstrated that the direct down-regulation of PKCe phenocopies SOD2 inhibition in acute myeloid leukemia cells and induces a marked increase in ROS [51].

ROS were tested 72 h after C2C12 transfection, when PKCe modulation was most evident. We found that ROS were significantly increased in C2C12 cells transfected with siPKCe as compared to siCTRL. The same effect on ROS was obtained when transfecting C2C12 with specific siRNA against SOD2 (Figure 5A–C). In contrast, SOD2 silencing did not affect mRNA expression and protein content of either PKCe or Myogenin (Figure 5D,E). These data suggest that PKCe could be an up-stream regulator of SOD2 expression. It is known that PKCe and other PKC isoforms are able to activate the transcription factor Nrf2, which in turn increases the antioxidant enzyme transcription program [62,63]. In order to evaluate the interaction between PKCe and Nrf2, we performed co-immunoprecipitation assays in C2C12 cells. As shown in Figure 5F, Nrf2 was detected in the PKCe pull-down (IP:PKCe) and, similarly, PKCe was detected in the Nrf2 pull-down (IP:Nrf2), thus indicating that PKCe interacts with Nrf2 in C2C12. 

## 4. Discussion

PKCe plays a pivotal role in cell proliferation, apoptosis, and differentiation [64,65]. We extensively investigated the role of PKCe in cell commitment, highlighting tissue-specific effects [37,38,39,40,41,42,43,44,45,46,47,48,49]. Indeed, we reported that high levels of PKCe impair intestinal progenitor cell differentiation [39], and that this kinase negatively regulates vessel progenitors differentiation in vitro and it reduces the levels of vascular differentiation markers in vivo [43]. In human hematopoiesis, while high levels of PKCe prevent erythroblast apoptosis [37], kinase down-regulation is required for mature platelet production [38,39,40]. Additionally, in muscle tissue, we demonstrated that PKCe sustains cardiac differentiation, as well as skeletal muscle regeneration [35,41]. 

Cell commitment and differentiation are also affected by ROS, whose effects may vary according to the intensity of stimuli and the cellular context. For instance, in the bone marrow, low levels of ROS are required for maintaining hematopoietic stem cell (HSC) self-renewal, while high levels (due to stress and inflammation) induce HSC differentiation and enhance cell motility [66]. Concerning skeletal muscle, fluctuations in intracellular ROS during myogenic differentiation are postulated to be one of the myogenesis-regulating factors [21]. Indeed, most of the literature shows that while intracellular ROS contribute to initiate skeletal muscle regeneration, continuous exposure to ROS at a high concentration may overwhelm the antioxidative capacity of cells, thereby exerting a negative effect on cell differentiation [1]. 

In our models, we observed a fast and transient ROS increase after only 24 h of differentiation, followed by a rapid decrease in more differentiated cells, supporting the concept that a higher number of ROS is necessary to start myogenesis [8,17,18,67], but subsequent antioxidant signaling activation is required to prevent the detrimental effects of ROS accumulation and allow cell differentiation. We demonstrated that the expression of SOD2 is significantly higher in myotubes as compared to myoblasts, suggesting that an increase in an antioxidant enzyme, in the late phase of in vitro differentiation, contributes to maintain low ROS in myotubes, likely preventing oxidative stress in myofibers, as observed in a pathological condition [1].

It is known that specific PKC isoforms may represent both a target of ROS as well as an up-stream regulator of redox signaling [54]. In non-muscle cells, it has been demonstrated that PKC alpha and epsilon sustain SOD2 expression, contributing to oxidative stress control [60,61]. Our data highlight a novel tissue-specific PKCe–redox axis in skeletal muscle. We proved that while ROS reduction induced by SOD2 silencing did not affect PKCe protein content, a forced in vitro PKCe down-regulation was able to induce a reduction in SOD2 protein content coupled with a significant increase in ROS. This is supported by the observation in vivo that PKCe^−/−^ mouse SOD2 protein content was dramatically reduced in skeletal myofibers.

We also identified a plausible molecular link between PKCe and SOD2 represented by Nrf2, a well-known SOD2 activator, by demonstrating the formation of the PKCe-Nrf2 complex in C2C12 cells. 

Overall, our study adds a novel element in the current understanding of ROS biology in skeletal muscle, demonstrating that PKCe promotes SOD2 expression likely via Nrf2, which in turn leads to a reduction in ROS, which is a crucial event to prevent oxidative damage in the later stages of myogenic differentiation.

## Figures and Tables

**Figure 1 cells-12-01792-f001:**
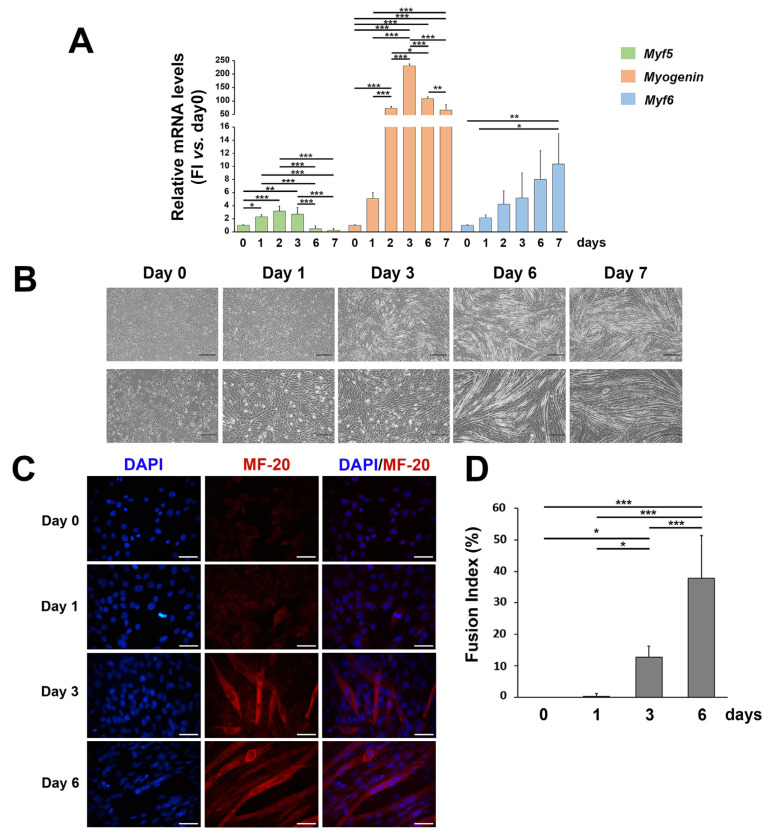
Morphological and molecular features of C2C12 cells during myogenic differentiation. (**A**) mRNA expression of *Myf5*, *Myogenin*, and *Myf6* in C2C12 cells cultured in the differentiation medium for up to 7 days. Results are representative of three independent experiments; values are reported as the fold increase in proliferating myoblasts (day 0) ± standard deviation. (Horizontal black bars represent *p* < 0.05 with the one-way ANOVA followed by the Tukey test). (**B**) Phase contrast images (at 5× and 10× magnification) representing morphological changes in C2C12 cells cultured in the differentiation medium for up to 7 days. (**C**) Representative immunofluorescence of proliferating C2C12 (day 0) and during C2C12 culturing in the differentiation medium (day 1, day 3, and day 6). Nuclei were stained in blue by DAPI, and the myosin heavy chain, composing the myofibril, in red by MF-20. The scale bar corresponds to 50 μm. (**D**) The fusion index analysis (ratio of the number of nuclei in the MF-20-positive myotubes as compared to the total number of nuclei). Reported values are expressed as a percentage (0% = no detectable fusion event among MF-20-positive cells). (* *p* < 0.05; ** *p* < 0.01, *** *p* < 0.001 with the one-way ANOVA followed by the Tukey test).

**Figure 2 cells-12-01792-f002:**
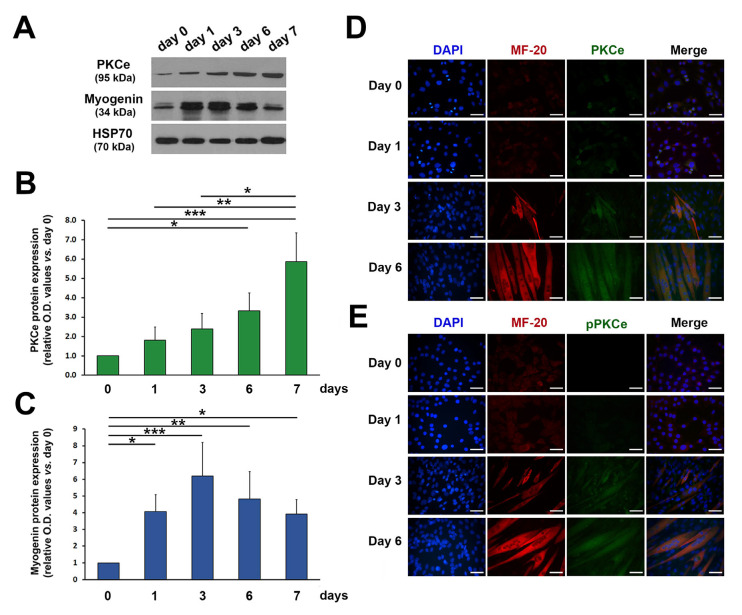
PKCe and phospho-PKCe (pPKCe) protein content in C2C12 cells during differentiation. (**A**) A representative Western blot showing protein content of PKCe and Myogenin during C2C12 cell differentiation; HSP70 was used as a loading control. (**B**,**C**) The densitometry analysis relative to PKCe/HSP70 (**B**) and Myogenin/HPS70 (**C**) protein content during C2C12 cell differentiation expressed as the fold increase in proliferating myoblasts (day 0) for each time point. Data are presented as means ± SD of at least 3 independent experiments. (* *p* < 0.05, ** *p* < 0.01, *** *p* < 0.001 with the one-way ANOVA followed by the Tukey test). (**D**,**E**) Representative immunofluorescence of PKCe and the myosin heavy chain (MF-20) (**D**) and of pPKCe and the myosin heavy chain (**E**) in the proliferating myoblast (day 0) and in the differentiation medium (day 1, day 3, and day 6). Nuclei were stained in blue by DAPI, myosin in red, and PKCe and pPKCe in green by specific antibodies (**D**,**E**), respectively. The scale bar corresponds to 50 μm.

**Figure 3 cells-12-01792-f003:**
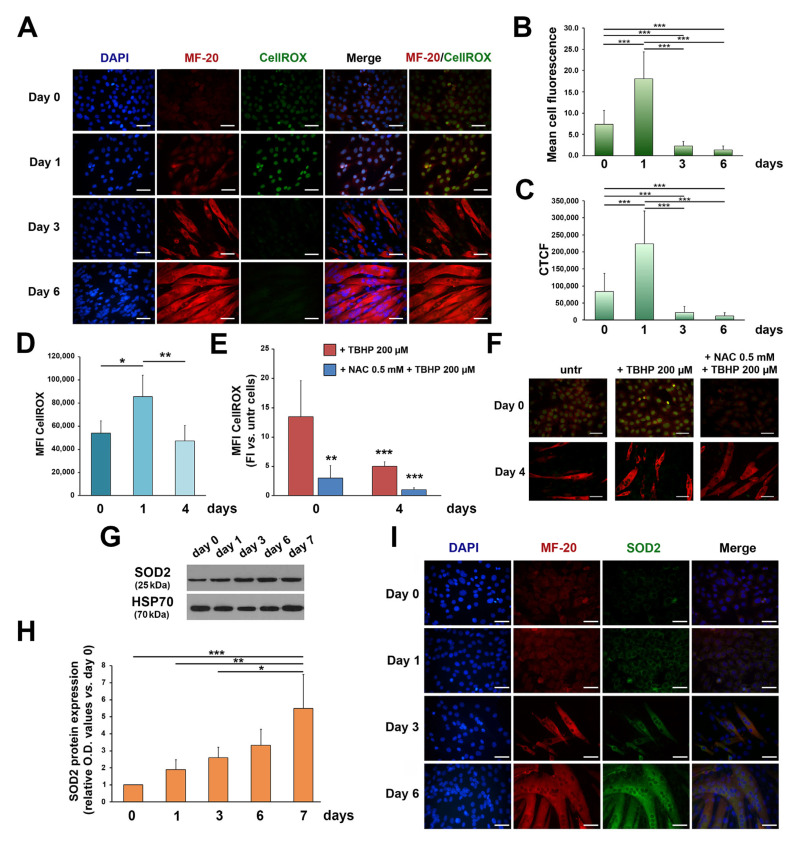
ROS and SOD2 protein content in C2C12 cells during differentiation. (**A**) Representative immunofluorescence images showing ROS in proliferating myoblasts (day 0) and during C2C12 culturing in the differentiation medium (day 1, day 3, and day 6). Nuclei are stained in blue by DAPI, the myosin heavy chain in red by MF-20, and ROS in green by CellROX staining. The scale bar corresponds to 50 μm. (**B**,**C**) The semiquantitative analysis of the immunofluorescent ROS signal using the mean cell fluorescence intensity quantified by ImageJ software (**B**) and the corrected total cell fluorescence (CTCF) (**C**) (*** *p* < 0.001 with the one-way ANOVA followed by the Tukey test). (**D**) The FCM analysis of ROS in proliferating myoblasts (day 0) and during C2C12 culturing in the differentiation medium (day 1 and day 4). Data are shown as the mean of mean fluorescence intensity (MFI) ± SD of three independent experiments (* *p* < 0.05, ** *p* < 0.01, the one-way ANOVA followed by the Tukey test). (**E**) The FCM analysis of ROS in proliferating myoblasts (day 0) and in differentiating myoblasts (day 4) treated with TBHP (+TBHP 200 µM) and with NAC and TBHP (+NAC 0.5 mM + TBHP 200 µM). ROS in TBHP- and NAC + TBHP-treated cells were normalized to untreated cells at each time point. Data are shown as the mean of MFI ± SD of three independent experiments (** *p* < 0.01, *** *p* < 0.001 with the one-way ANOVA followed by the Tukey test). (**F**) Representative immunofluorescence images showing ROS in proliferating myoblasts (day 0) and in differentiating myoblasts (day 4) treated with TBHP (+TBHP 200 µM) and with NAC and TBHP (+NAC 0.5 mM + TBHP 200 µM). The myosin heavy chain, composing the myofibril, is shown in red by MF-20 staining, and ROS in green by CellROX staining. The scale bar corresponds to 50 μm. (**G**) A representative Western blot showing SOD2 protein content during C2C12 cell differentiation; HSP70 was used as a loading control. (**H**) The densitometry analysis of SOD2 protein content during C2C12 cell differentiation. Protein expression was normalized to HSP70 protein content for each time point and data are reported as the fold increase in proliferating myoblasts (day 0) for each time point. Data are presented as means ± SD of at least 3 independent experiments. (* *p* < 0.05, ** *p* < 0.01, *** *p* < 0.001 with the one-way ANOVA followed by the Tukey test). (**I**) Representative immunofluorescence of SOD2 and MF-20 protein content in proliferating myoblasts (day 0) and during C2C12 culturing in the differentiation medium (day 1, day 3, and day 6). Nuclei are stained in blue by DAPI, myosin in red by MF-20, and SOD2 in green. The scale bar corresponds to 50 μm.

**Figure 4 cells-12-01792-f004:**
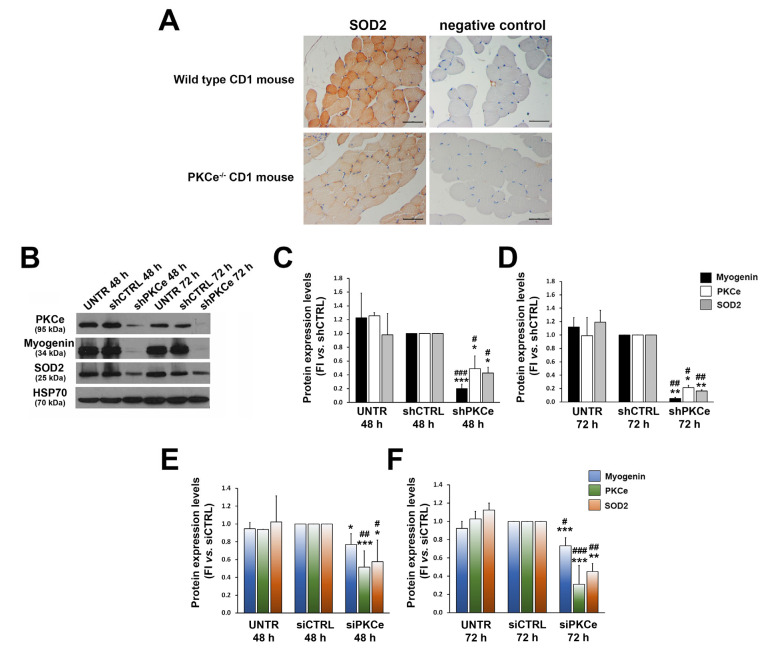
Effects of PKCe down-modulation on SOD2 expression. (**A**) Representative immunohistochemistry of SOD2 protein content in tibialis muscle of wild-type and PKCe^−/−^ mice. The scale bar corresponds to 50 μm. (**B**) A representative Western blot showing PKCe, Myogenin, and SOD2 protein content in C2C12 untreated (UNTR) and infected with PKCe shRNA (shPKCe) or a shRNA negative control (shCTRL) after 48 and 72 h of puromycin selection. HSP70 was used as a loading control. (**C**,**D**) The densitometry analysis of PKCe, Myogenin, and SOD2 protein content in C2C12 untreated (UNTR) and infected with PKCe shRNA (shPKCe) or a shRNA negative control (shCTRL) after 48 (**C**) and 72 h (**D**) of puromycin selection. Protein content was normalized to HSP70 protein content for each experimental condition and each time point. Data are reported as the fold increase in UNTR and shPKCe to shCTRL. Data are presented as means ± SD of at least 3 independent experiments (* vs. shCTRL: * *p* < 0.05, ** *p* < 0.01, *** *p* < 0.001; # vs. UNTR: # *p* < 0.05, ## *p* < 0.01, ### *p* < 0.001, the one-way ANOVA followed by the Tukey test). (**E**,**F**) The densitometry analysis of PKCe, Myogenin, and SOD2 protein content in C2C12 untreated (UNTR) and transfected with PKCe siRNA (siPKCe) or a siRNA negative control (siCTRL) after 48 (**E**) and 72 h (**F**) of transfection. Protein content was normalized to HSP70 for each experimental condition and each time point. Data are reported as the fold increase in UNTR and siPKCe to siCTRL. Data are presented as means ± SD of at least 3 independent experiments (* vs. siCTRL: * *p* < 0.05, ** *p* < 0.01, *** *p* < 0.001; # vs. UNTR: # *p* < 0.05, ## *p* < 0.01, ### *p* < 0.001, the one-way ANOVA followed by the Tukey test).

**Figure 5 cells-12-01792-f005:**
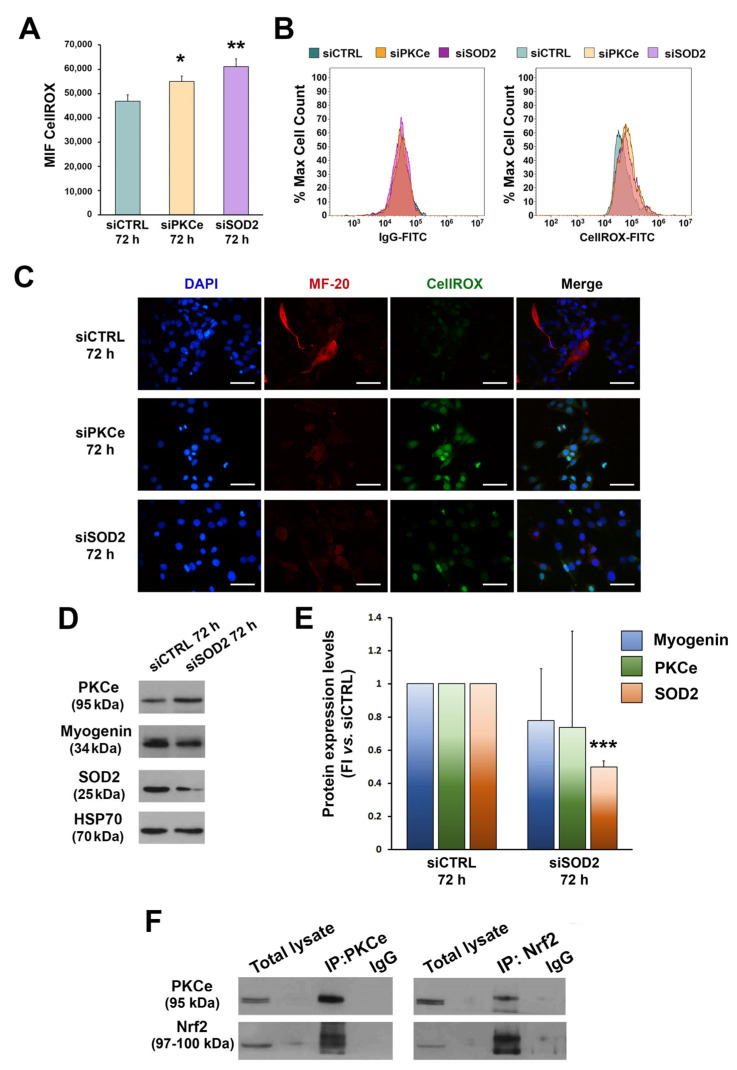
Effects of PKCe and SOD2 down-modulation on ROS. (**A**) The FCM analysis of ROS in C2C12 transfected with PKCe siRNA (siPKCe), SOD2 siRNA (siSOD2), or a control siRNA (siCTRL) after 72 h of transfection. Data are shown as the mean of mean fluorescence intensity (MFI) ± SD of three independent experiments. (* vs. siCTRL: * *p* < 0.05, ** *p* < 0.01, the one-way ANOVA followed by the Tukey test). (**B**) Representative histograms of the FCM analysis of ROS. (**C**) Representative immunofluorescence of ROS in C2C12 transfected with PKCe siRNA (siPKCe), SOD2 siRNA (siSOD2), or a siRNA negative control (siCTRL) after 72 h of transfection. Nuclei are shown in blue by DAPI staining, the myosin heavy chain in red by MF-20 staining, and ROS in green by CellROX staining. The scale bar corresponds to 50 μm. (**D**) A representative Western blot showing PKCe, Myogenin, and SOD2 protein content in C2C12 transfected with SOD2 siRNA (siSOD2) or a siRNA control (siCTRL) after 72 h of transfection. HSP70 was used as a loading control. (**E**) The densitometry analysis of PKCe, Myogenin, and SOD2 protein content in C2C12 transfected with SOD2 siRNA (siSOD2) or a siRNA control (siCTRL) after 72 h of transfection. Protein content was normalized to HSP70 for each experimental condition. Data are reported as the fold increase in siSOD2 to siCTRL. Data are presented as means ± SD of at least 3 independent experiments (*** *p* < 0.001, the one-way ANOVA followed by the Tukey test). (**F**) A representative Western blot showing PKCe and Nrf2 protein interaction in C2C12. C2C12 lysates were subjected to immunoprecipitation with a specific anti-PKCe antibody and anti-Nrf2 antibody (IP-PKCe and IP-NRF2), and proteins detected with Western blotting as indicated. Samples were blotted for PKCe and Nrf2.

**Table 1 cells-12-01792-t001:** Sequences of the primer pairs employed for RT-qPCR analysis.

Gene/Protein Name NCBI Reference Sequence Identifier	Forward Primer	Reverse Primer
*Mrf4* (*Myf6*) (NM_008657.3)	GAGATTCTGCGGAGTGCCAT	TTCTTGCTTGGGTTTGTAGC
*Myog* (NM_031189.2)	ATCCAGTACATTGAGCGCCT	GCAAATGATCTCCTGGGTTG
*Myf5* (NM_008656.5)	TGAGGGAACAGGTGGAGAAC	AGCTGGACACGGAGCTTTTA
*Gusb* (NM_001357025.1)	CCGCTGAGAGTAATCGGAAAC	TCTCGCAAAATAAAGGCCG

## Data Availability

The data presented in this study are available on request from the corresponding author.

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
