# Peer review of "Interplay between Protein Kinase C Epsilon and Reactive Oxygen Species during Myogenic Differentiation"

_cells, 2023, doi:10.3390/cells12131792_

Round 1
Reviewer 1 Report
Thank you for the opportunity to review this manuscript. This paper is highly interesting and contributes significantly to our understanding of the biological role that reactive oxygen species (ROS) play in skeletal muscle. The authors report, that PKCe may promote SOD2 protein content, possibly via Nrf2, which subsequently leads to a reduction of ROS generation. The authors suggest that this is a crucial event in preventing oxidative damage during the later stages of myogenic differentiation. The manuscript presents valuable insights into the interplay between PKCe, SOD2, and ROS in skeletal muscle biology.
The authors have prepared an acceptable manuscript. However, I have some concerns existing in the manuscript.
Major:
I have looked many times at Figure 2A and 3G (WB) and have the impression that line bands are the same for the protein content of HSP70 (Fig. 2A PKCe and Myogenin; Fig. 3G SOD2), which was used for both different blots. Could you explain it, please?
Page 9, lines 303-306: I would like to kindly inquire about the rationale behind choosing SOD2 over SOD1 or GPx, or possibly both. Firstly, SOD2 (MnSOD) is primarily located in mitochondria, and secondly, the influx of manganese (Mn) and the dynamic expression of Mn-transporting proteins are likely crucial factors in skeletal muscle cell differentiation. Have you considered measuring the major Mn transporters in cells or TA? It would be helpful if you could provide further clarification on this matter. Additionally, I recommend shifting the focus related to SOD2 and including additional explanations in the paper.
Page 11: TA muscle primarily consists of white fibers (fast-twitch, glycolytic) in mice, so I am curious why it was used instead of the soleus muscle, which primarily contains red fibers (slow-twitch, oxidative, mitochondria), and thus potentially exhibits higher activity or protein content of SOD2. It seems rational to use both muscles. Please explain this aspect.
Minor:
I would also suggest maintaining consistency in the use of terminology throughout the manuscript. For example, employing consistent terms such as ROS (where generation refers to oxidative damage and production rather to physiological conditions) instead of ROS levels, ROS production, ROS generation. Similarly, for protein-related discussions, maintaining consistency in the use of terms like protein content instead of alternative terms (expression, levels), and RNA expression (instead of using another term).
Furthermore, there appear to be some editing mistakes in the manuscript. I recommend thoroughly reviewing the entire manuscript and making appropriate corrections as necessary.
The quality of the English language is OK.
Reviewer 2 Report
The authors highlighted the role of PKCe in the activation of SOD2 via Nrf2, and the consequent reduction of ROS that promotes myogenic differentiation. The work is well structured, is comprehensive and shows in vitro and in vivo experiments performed on wild type mice with PKCe-Knock out.
However, some revisions are necessary.
The authors measured the levels of intracellular ROS; because the ROS are also produced in cytosol where O-°2 is converted by CuZn,Superoxide dismutase (SOD1) in Hydrogen peroxide, should by interessed investigate the ROS production in mitochondria using a specific probe such as MitoSOX.
In all figures, the labels should be written in a larger font.
In figure 2 the analysis of WB (panel A) is not complete, in particular pPKCe present in panel E is missing, in addition the histogram showing the activation of PKCe (pPKCe/PKCe) should be added.
In figure 4 the authors shows the Effects of PKCe down-modulation of SOD2 expression in C2C12 cells by infection with PKCe shRNA (shPKCe) and siRNA (siPKCe) in the E and F panels; howewer PKCe, myogenin and SOD2 protein levels in C2C12 cells untreated (UNTR) are missing.
In the text should be specified that SOD2 has a mitochondria localization.
A short comment on superoxide dismutase isoenzymes could be useful.
Round 2
Reviewer 1 Report
The explanation provided and the corrections incorporated in the manuscript are satisfactory.
Reviewer 2 Report
Authors of the manuscript: "Interplay between Protein Kinase C epsilon and reactive oxygen species during myogenic differentiation" answered my questions, so the manuscript can be published